# The Aggravating Role of Failing Neuropeptide Networks in the Development of Sporadic Alzheimer’s Disease

**DOI:** 10.3390/ijms252313086

**Published:** 2024-12-05

**Authors:** Miklós Jászberényi, Balázs Thurzó, Arumugam R. Jayakumar, Andrew V. Schally

**Affiliations:** 1Department of Pathophysiology, University of Szeged, P.O. Box 427, H-6701 Szeged, Hungary; thurzo.balazs@med.u-szeged.hu; 2Emergency Patient Care Unit, Albert Szent-Györgyi Health Centre, University of Szeged, Semmelweis u. 6, H-6725 Szeged, Hungary; 3Department of Obstetrics, Gynecology and Reproductive Sciences, University of Miami Miller School of Medicine, Miami, FL 33136, USA; ajayakumar@med.miami.edu (A.R.J.); aschally@med.miami.edu (A.V.S.)

**Keywords:** Alzheimer’s disease, neuropeptide, ghrelin, apelin, RFamides, neuromedin, orexin, urocortin, CRF, GHRH, GnRH

## Abstract

Alzheimer’s disease imposes an increasing burden on aging Western societies. The disorder most frequently appears in its sporadic form, which can be caused by environmental and polygenic factors or monogenic conditions of incomplete penetrance. According to the authors, in the majority of cases, Alzheimer’s disease represents an aggravated form of the natural aging of the central nervous system. It can be characterized by the decreased elimination of amyloid β_1–42_ and the concomitant accumulation of degradation-resistant amyloid plaques. In the present paper, the dysfunction of neuropeptide regulators, which contributes to the pathophysiologic acceleration of senile dementia, is reviewed. However, in the present review, exclusively those neuropeptides or neuropeptide families are scrutinized, and the authors’ investigations into their physiologic and pathophysiologic activities have made significant contributions to the literature. Therefore, the pathophysiologic role of orexins, neuromedins, RFamides, corticotrope-releasing hormone family, growth hormone-releasing hormone, gonadotropin-releasing hormone, ghrelin, apelin, and natriuretic peptides are discussed in detail. Finally, the therapeutic potential of neuropeptide antagonists and agonists in the inhibition of disease progression is discussed here.

## 1. Sporadic Alzheimer’s Disease as an Accelerated Form of the Aging of the Central Nervous System (CNS)

Alzheimer’s disease is the most frequent form of the proteinopathies of the central nervous system (CNS), in which various undigestible protein derivatives accumulate in the brain and cause severe neurodegeneration. These proteinopathies include the prion diseases (PrDs) (e.g., Creutzfeldt–Jakob disease), the synucleinopathies (e.g., Parkinson’s disease), the amyloidoses (e.g., Alzheimer’s disease, early stage), and the tauopathies (e.g., amyotrophic lateral sclerosis (ALS) and Alzheimer’s disease, late stage). In some disorders, rarer protein types (e.g., huntingtin, ataxin, and superoxide dismutase (SOD)) deposit first in specific CNS regions (in the striatum, the prefrontal cortex, the spinal cord, and the cerebellum) and then in the whole CNS. Ultimately, they manifest themselves as Huntington’s disease, ALS, frontotemporal lobar degeneration (FTD), and spinocerebellar ataxia, respectively [1,2].

In these diseases, monomeric proteins with an α-chain structure misfold, gain β-pleat conformation, and aggregate [1]. The formation of the β-pleat variant can be attributed to three factors, which also determine the main etiologic groups of the disorders. In the congenital group, which represents the smallest portion of familial cases, the β-pleat is formed due to monogenic mutations of complete penetrance. The onset of the disease is markedly early, and the progression is relentlessly fast. They all represent familial Alzheimer’s disease (FAD) with the clinical appearance of early-onset Alzheimer’s disease (EOAD). Personality changes, anxiety, mood swings, memory loss, impaired thinking, blurred vision, difficulty speaking, sudden and jerky movements, and increased startle response are the common initial symptoms [3,4]. Then, complete mental deterioration develops in months and the patient succumbs to death in less than a year. However, monogenic conditions of incomplete penetrance, such as certain apoE polymorphisms, may give rise to late-onset Alzheimer’s disease (LOAD) [5,6].

In sporadic Alzheimer’s disease (SAD), a specific constellation of polygenic and environmental factors is required to facilitate β-pleat formation. This combination sets off a pathophysiologic chain reaction, leading to untimely cognitive decline. This cascade process is attributed to a biochemical blind alley; although both the α-chain and the β-pleat variants are stable thermodynamically, the free energy level of the β-pleat conformation is lower and it is also resistant to proteases [7]. Therefore, if the α-chain variant is facilitated “to jump over” the energy barrier, the β-pleat isomer inevitably accumulates over time. This conformational change can be accelerated by several factors [7]. The necessary momentum to bypass the energy barrier can be provided by kinetic impulses such as repeated cerebral contusions and concussions from traumatic brain injuries (TBIs) giving rise to chronic traumatic encephalopathy (CTE) [8,9]. A gel–sol shift in the cytoplasm may also speed up these conformational changes [7]. Organic solvents (e.g., general anesthetics and organic solvents in glue) and head traumas are the most frequent reasons for such a gel–sol shift. But general anesthetics (especially smaller-sized volatile agents) can directly initiate misfolding and oligomerization giving rise to the clinical syndrome of postoperative cognitive dysfunction (POCD) [10]. Ultimately, metallic ions [(aluminum (Al^3+^), manganese (Mn^2+^), and copper (Cu^+^)] and acquired β-pleat isoforms can act as catalysts, reducing the height of the energy barrier [7]. From then onward, the β-pleat templates operate as autocatalysts for misfolding and the initial foci later stimulate propagation in a vicious cycle [7]. The progression of these secondary forms is more insidious, and the symptoms become apparent much later, which makes a differential diagnosis more complicated, especially the separation from chronic vascular dementia based upon clinical findings.

The autocatalytic property of the β-pleat templates is also responsible for the contagious nature of some variants of these diseases [11,12]. Ingestion or inoculation of β-pleat templates is accountable for nidus formation and propagation of the pathobiochemical conformation in the infectious group. These proteinaceous, infectious particles were named prions, and later, they were verified to represent the pathophysiologic β-pleat variant (PrP^sc^) of α-helical chaperon molecules (PrP^c^) [13]. That is why prions were once considered unique transmissible agents, but recent progress in the research has revealed that the β-pleat variants of several common proteins (α-synuclein, amyloid, tau protein, huntingtin, ataxin, and SOD1) share the pathophysiologic properties of prions, such as protease resistance, catalytic template activity, propensity to aggregation, and infection-like spread [1]. Similar to the monogenic forms, the infectious form progresses dramatically fast.

The spread of protein misfolding operates at different levels during the pathogenesis of neurodegenerative disorders [2]. It includes intermolecular, cell-to-cell, and brain region-to-brain region stages. As mentioned above, misfolding can also spread from individual to individual as has been demonstrated in the case of PrDs. Contagious spread is usually related to inoculation of contaminated human tissues (extracted growth hormone (GH), corneal transplant, and dural graft), deep cortical EEG procedure, or consumption of infected proteins, such as kuru or bovine spongiform encephalopathy (BSE) and the variant of human Creutzfeldt–Jacob disease (vCJD)] caused by scrapie [14]. In addition, vCJD infectious spread has already been suspected in Parkinson’s disease, Alzheimer’s disease, and ALS [15]. This is why a new concept is being elaborated on, which assigns a general etiopathogenetic role to protease-resistant catalytic β-pleat conformations [16].

In the case of Alzheimer’s disease, several etiological factors have been implicated in its pathogenesis (Figure 1) [17,18].

It was mentioned that epidemiologically, Alzheimer’s disease exists in familial and sporadic forms. FAD represents 20% of the cases, and it clinically may appear as LOAD or the most-feared EOAD. LOAD cases of FAD usually can be attributed to monogenic conditions of incomplete penetrance, such as apoE polymorphism, and they account for circa 15–20% of the cases. EOAD represents only a small fraction (1–5%) of the cases. However, due to its complete penetrance, it develops below 65 years and kills a relatively young patient twice, taking the soul first. The genetic background of EOAD is well known, and we have several animal models to examine it [19,20,21]. This variant is transmitted in an autosomal-dominant manner and results from increased formation of amyloid-β_1–40_ and amyloid-β_1–42_, a process which results from a pathobiochemical metabolism of amyloid precursor protein (APP) [17,18].

Normally, APP is sequentially cleft by α- and γ-secretases and the process yields soluble APP-α (SAPP-α) [17]. Actually, the α-secretase name refers to a whole group of enzymes, which belong to the ADAM (“a disintegrin and metalloprotease domain”) group (ADAM9, ADAM10, ADAM17, and ADAM19) of peptidases. The α-secretases, in addition to cleavage of APP, cut further important biologically active peptides such as tumor necrosis factor-α (TNF-α; cleft by ADAM17), L-selectin (cut by ADAM17), cadherins (e.g., ephrin processed by ADAM10), the triggering receptor expressed on myeloid cells 2 (TREM2, cut by ADAM10 and ADAM17), and Notch peptide (cleft by ADAM10 or ADAM17). ADAM17 is also called tumor necrosis factor-α-converting enzyme (TACE) [22,23,24]. SAPP-α represents the normal, physiologic variant. It is neuroprotective and regulates axonal transport, synapse formation, synaptic repair, and neural plasticity [25,26,27]. Equally important is that the tandem activity of ADAMs and the γ-secretase is also required for the proper signaling in the Notch pathway [28]. Similar to the processing of SAPP-α, cleavage of the Notch protein by an ADAM is followed by γ-secretase-mediated proteolysis. According to experimental data, the Notch pathway plays an essential role in neurogenesis and synaptic and dendritic remodeling even in the adult brain. That is why mutation or oversaturation of γ-secretase can also contribute to the development of Alzheimer’s disease through the failure of Notch signaling [29,30]. Perhaps that is the reason for the failure of clinical trials with γ-secretase inhibitors, which repeatedly returned disappointing results [29]. In contrast, in pathophysiologic circumstances, an abnormal cleavage of APP by the β-site APP-cleaving enzyme-1 (BACE1) happens prior to the action of γ-secretase [17]. BACEs normally cleave voltage-gated sodium channels, neuregulin, and pre-melanosome protein [31,32]. However, in the processing of APP, BACE1 activity results in serious neurotoxic amyloid-β (Aβ) derivatives [33]. Recently, several other enzymes such as meprin-β, cathepsin B, and ADAMTS4 (a disintegrin and metalloproteinase with a thrombospondin type 1 motif, member 4) have been identified as alternative β-secretases [34].

In EOAD, several dominantly inherited abnormalities have been discovered as causative factors [17,35,36]. The APP is coded by chr. 21; therefore, sufferers of Down syndrome have an extra copy of the APP. This way, unavoidably, in their CNS, the overproduction of Aβ can be observed and the disease develops in patients when they are in their early forties. Another reason is when APP mutations render the APP inaccessible to α-secretases, APP can be processed only by BACEs. Regarding the γ-secretase activity, two rival hypotheses have gained popularity among researchers. Supporters of the gain-of-function (GOF) mutation hypothesis argue that the GOFs that affect the catalytic presenilin (PSEN) sub-domains of the γ-secretase lead to acceleration in both the physiologic and the pathophysiologic pathways. Obviously, over time, the insoluble, oligomers of Aβ derivatives (Aβ_1–40_ and especially the even more toxic Aβ_1–42_) [37] will inevitably accumulate at the expense of the soluble, decomposable conformation [38]. On the other hand, those who measured decreased activity of the γ-secretase, suggest that loss-of-function mutations (LOF) are responsible for this form of FAD. In their view, not the Aβ aggregation but decreased Notch signaling must be blamed as the culprit [29,30]. That is why they dismiss the concept of γ-secretase inhibition [29]. Theoretically, resistance to α-secretase should represent a more serious abnormality of APP metabolism than the GOF mutations of the PSEN, if an exclusive pathophysiologic role is attributed to amyloid-β overproduction and oligomer burden. It is because GOF mutations of PSEN increase both the physiologic SAPP-α and the toxic Aβ production, while APP mutations at α-secretase cleavage points simply exclude SAPP-α formation and allow only Aβ secretion. However, several lines of evidence suggest that PSEN mutations have more severe consequences than those phenotypes with α-secretase resistance [39,40]. This finding therefore supports the view that the failure of Notch signaling, caused by LOFs of PSENs, is a more important causative factor in those monogenic forms of Alzheimer’s disease, which are brought about by γ-secretase mutations. Increased cleavage of APP at the β-site by BACE1 represents the last known monogenic forms of FAD. These mutations make the APP more accessible to BACE1, and the Aβ processing is accelerated by 10- to 50-fold. To a lesser extent, epigenetic upregulation of BACE1 expression can lead to similar consequences [33]. Aβ production increases and pathobiochemical alterations may oversaturate the γ-secretases. This disrupts the Notch pathway, seriously compromising the process of normal neurogenesis [29,30].

In the past few decades, studying EOAD has greatly improved our understanding of the underlying pathophysiological processes of Alzheimer’s disease. However, these experiments shed light only on the consequences of Aβ and hyperphosphorylated tau accumulation. We could hardly learn anything from them about the prevention and management of the sporadic form of Alzheimer’s disease (SAD), which represents the lion’s share of the cases (75–85%) and usually manifests itself clinically in late-onset Alzheimer’s disease (LOAD) [19,20,21]. LOAD accounts for 95–99% of the cases [19,20] and can be attributed to a few monogenic and numerous polygenic and environmental factors [38]. The penetrance of LOAD is far less than that of EOAD since familial clustering is 3-fold higher in EOAD than in LOAD [19,20]. Further, while in EOAD the increase in Aβ production is the characteristic alteration, LOAD predominantly reflects the abnormality of the opposite end of Aβ metabolism. Namely, not the formation, but the decomposition is affected by environmental factors or several minor mutations which aggravate each other’s consequences.

In the cell, Aβ can either be broken down by microglial cells (phagocytosis) or by neural elements. In the latter, astrocytes take up Aβ by receptor-mediated processes and try to decompose it intracellularly [17]. The breakdown of Aβ is a cumbersome process. Intracellularly, it happens in the endosomes, lysosomes, and proteasomes after Aβ is taken up and transported by multivesicular bodies to the place of degradation. Extracellularly, Aβ is cleared by ubiquitous peptidases, with different efficiency. The most effective enzymes are angiotensin-converting enzyme (ACE), endothelin-converting enzyme (ECE), insulin-degrading enzyme (IDE), neprilysin (NEP), several matrix metalloproteases (MMPs), and plasmin [17,41,42].

As outlined above, in LOAD, the clearance process can be impaired by monogenic mutations of incomplete penetrance and polygenic and environmental factors. Obviously, in several conditions, the contribution of individual risk factors cannot be exactly quantified, because in the pathophysiologic processes, their impact is completely interwoven. Other than age, more than 60 factors have been suggested so far [43]: with the most prevalent that have been connected to the disease being repeated head injuries, lipid metabolism disorders, type II diabetes mellitus (type II DM), vascular diseases (e.g., hypertension), stressful conditions (e.g., major depression), chronic inflammatory diseases, female gender and menopause, and disruption of the glymphatic circulation [19,20,43,44,45,46].

The most important monogenic factor of incomplete penetrance, which has been indirectly linked to LOAD, is apolipoprotein E4 (apoE4) hetero- and homozygosity. However, carrying such a mutation does not necessarily mean a “death sentence”. In general, apoE4 carriers are several (three for heterozygotes and six for homozygotes) times more prone to develop the disease, while apoE2 homozygosity represents a protective phenotype [5,6]. In the CNS, the most significant receptor group for apoEs is the low-density lipoprotein receptor-related proteins (LRPs). LRPs are responsible for cholesterol and Aβ uptake into the neurons and astrocytes. Since apoE4 shows the highest and apoE2 confers the lowest affinity to their receptors, the apoE4-Aβ complex increases the receptor-mediated uptake of Aβ into the neural elements and leads to intracellular aggregation [5,6,17]. Further, apoE4 was demonstrated to disrupt Aβ clearance at the blood–brain barrier (BBB) in an isoform-dependent manner (ApoE4 > ApoE3 > ApoE2) and to promote cerebral amyloid angiopathy pathogenesis [47].

Polygenic conditions, metabolic syndrome, and type II DM appear to be the most important risk factors [48]. Nevertheless, the exact pathophysiological mechanism behind this epidemiologic association has not yet been clarified. It seems that at the beginning of type II DM, which is characterized by hyperinsulinemia, oversaturation of IDE, and a decrease in Aβ clearance can be a significant contributing factor [49,50]. It seems highly probable since IDE is one of the most important enzymes, which are responsible for the breakdown of Aβ. However, non-enzymatic glycation of Aβ and increased neuronal uptake by the receptors for advanced glycation end products (RAGE) can also aggravate the pathophysiologic process [17]. Later, in the so-called spent-out phase, a decrease in insulin secretion and hypoglycemia of the insulin-dependent phagocytes will impair the microglial scavenge of Aβ, which precipitously increases the intracellular burden by aggregates. This ultimately causes neuronal inflammation and overproduction of acute-phase proteins such as α2-macroglobulin (α2MG). Like apoE, α2M functions as a chaperone and accelerates especially the neuronal uptake and intracellular aggregate formation of Aβ by the LRPs [17,38]. Through neuroinflammation [43,51,52], several infectious (e.g., herpes simplex infection) [53] or non-infectious (e.g., multiple sclerosis (MS) [54] and/or major depression [55]) diseases can make the brain more susceptible to developing amyloid rafts. Moreover, in chronic inflammatory processes, such as atherosclerosis, a common finding is the upregulation of plasminogen activator inhibitors (PAIs), which aggravate the development of “inflammatory sludge”. However, in this way, in the CNS, they can also inhibit the activity of plasmin and the removal of Aβ [56]. An important caveat of modern pharmacology is the inadvertent application of MMP inhibitors to delay the spread of chronic inflammatory or metastatic processes. Some of them (such as Periostat) [57] are already approved by the FDA, but their application raises concerns from theoretical points of view, since in the long term, they may also interfere with the degradation of Aβ.

In hypertension, ACE activity can be impaired by the condition itself or by therapy. Both in the primary and secondary forms of hypertension, those conditions which are associated with high renin activity may lead to oversaturation of ACE [58]. Since ACE2 is the specific receptor for COVID-19 binding [59], COVID-19 infection can also impair the proper degradation of Aβ by ACE2 [59,60].

Infections by neurotropic viruses, as triggering factors in the development of neurodegenerative disorders (e.g., post-encephalitic Parkinsonism after encephalitis lethargica caused by influenza) have already been suggested [61,62,63]. It appears especially important in those infections that are associated with anosmia, though some authors suggest that anosmia and neuroinvasion may reflect two independent processes [64]. Retrograde olfactory neuroinvasion is a common finding in COVID-19 [65], and this process was also described in the case of herpes and influenza virus infections [66] and may lead to direct neuronal loss due to the cytopathic action of the virus. This phenomenon invokes two important neurological parallels. First, practically in all forms of neurodegenerative disorders [most typically in Lewy body dementia, Parkinson’s disease, Alzheimer’s disease, FTD, and progressive supranuclear palsy (PSP)], anosmia can be an early warning sign of insidious disease progression [3,4,67]. Second, the olfactory cortex possesses unique connections to the amygdala and the hippocampus, and both regions are the most severely affected by neurodegeneration [68]. Therefore, it is intriguing to surmise that cell-to-cell propagation of the direct cytopathic viruses, or the secondary metabolic derangement through the manipulation of ACEs, can initiate the pathophysiological cascade of neurodegeneration. Not only infections but also inflammatory conditions with high levels of mediator release can contribute to Alzheimer’s disease. ACE dysfunction may also aggravate these processes, as ACE also functions as a degrading enzyme of such inflammatory mediators such as bradykinin and Substance P [69]. In inflammation, ACEs can also be oversaturated, and the breakdown of Aβ consequently decreases. This way, theoretically, even antihypertensive therapy could impair Aβ degradation, since ACE inhibitors belong to the most common group of antihypertensives. This especially goes for those promising antihypertensive combinations, which aim to inhibit both NEP and ACE in heart failure. NEP inhibition is beneficial since NEP breaks down natriuretic peptides (NPs). This way, in addition to the decrease in peripheral resistance (afterload) caused by ACE, the elimination of systemic edemas by NPs, through increased diuresis, will decrease the preload [70].

The glymphatic system is responsible for waste clearance in the CNS [44,45,46]. It is propelled by the heart, and its waves coincide with the pulse waves. A decline in cerebrospinal fluid (CSF) production, failure of perivascular aquaporin-4 (AQP4) polarization, and gliosis all directly impede directional glymphatic flow and debris removal. However, indirectly, other factors seem to play an equally important role. Increased atrial pressure (e.g., in atrial fibrillation, mitral valve diseases, and left ventricular failure), sleep deprivation, and decline in sleep quality (e.g., obstructive sleep apnea and substance abuse) give rise to the stagnation of glymphatic flow with diminished brain fluid clearance. In normal aging, all the aforementioned problems can be observed in patients [44,45,46]. Retention of extracellular protein debris facilitates protein aggregation in a vicious cycle. The spread strikingly follows the routes of the glymphatic circulation reflecting that its failure is a marked contributing factor to the progress of the disease [44,45,46].

Female gender and menopause [71,72], as well as stressful conditions and neuropsychiatric disorders [73,74,75,76], all appear to be very important epidemiologic factors. However, in their pathophysiology, neuropeptides are the lion’s share and their contribution will be discussed in the following chapters.

Among exclusively environmental contributors that can accelerate the progress of Alzheimer’s disease, the above-discussed viral infections [53], CTE [8,9], and intoxications received the most attention [10,77]. CTE caused by single or repeated TBI represents by far the most important causative factor. It directly impairs the especially vulnerable cholinergic [78,79] and monoaminergic afferents [80,81], which originate in the basal ganglia, the thalamus, and other regions of the limbic system and target the glutamatergic pyramidal cells of the hippocampus. Moreover, CTE can lead to inflammation and increased formation and aggregation of β-pleats. As for intoxications, organic solvents, heavy metals (lead, arsenic, cadmium, and mercury), Al^3+^, Mn^2+^, and Cu^+^ have been implicated in the pathogenesis [10,77]. Toxic metals have a well-known deleterious effect on astrocytes [82] and macrophages such as the microglial cells. These can inhibit phagocytotic removal and intracellular decomposition of Aβ and shift the activity of macrophages into a proinflammatory direction [83,84]. Moreover, toxic metallic ions can form pathologic chelates with the classic enzymes of the Aβ decomposition pathway, replacing the physiologic metallic cofactors such as calcium (Ca^2+^), magnesium (Mg^2+^), and zinc (Zn^2+^) [23,42,85,86]. This impairs the normal processing of APP, the signaling of the Notch pathway, and the decomposition of Aβ. It is because both the ADAMs and the breakdown enzymes (IDE, ACE, ECE, NEP, and MMPs) are all metalloproteases, [23,42,86]; therefore, their inhibition decreases both the generation of SAPP-α and the decomposition of Aβ.

At a cellular level, Aβ first accumulates extracellularly, and then it forms insoluble “rafts” in the cell membrane [17]. This damages the membrane-bound ATPases, stimulates the N-methyl-D-aspartate (NMDA) receptors, and initiates the formation of membrane attack complexes (MACs). This way, membrane conductivity increases, which results in cytosolic calcium Ca^2+^ overload. Extracellular Aβ also directly inhibits glucose transporters (GLUT1 and GLUT4). When Aβ gets into the synaptic cleft, it seriously interferes with the normal synaptic transduction of operative and trophic signals. Aβ can bind to glutamate (Glu) receptors such as the NMDA and the α-amino-3-hydroxy-5-methyl-4-isoxazole propionic acid (AMPA) receptor, as well as the nicotinic acetylcholine (Ach) receptor and the voltage-gated Ca^2+^ channel (VGCC). Therefore, it impairs synaptic transmission by altering the balance between long-term potentiation (LTP) and long-term depression (LTD). Aβ also binds to the p75 neurotrophin receptor (p75NTr) and tyrosin kinase A and B (TrkA and TrkB) receptors. This way, both brain-derived neurotrophic factor (BDNF) and nerve growth factor (NGF) secretion decreases [17]. In addition to inhibition, Aβ also causes endocytosis and the downregulation of the aforementioned receptors.

As discussed above, undigested toxic Aβ can primarily enter the neurons via the LRPs and RAGE receptors [17]. This receptor-mediated process is facilitated by chaperone molecules such as apoE4 and α2M. On the one hand, cellular uptake of Aβ to microglial cells and astrocytes results in their stimulation and the release of proinflammatory cytokines. On the other hand, in the neurons, this gives rise to intracellular aggregation, neuritic plaque formation, and neurotoxicity [17]. The neurotoxic process is centered around the pathophysiological cascades caused by mitochondrial injury [17]. Once in the cell, Aβ molecules will bind to the mitochondrial membrane and then get into the mitochondrion. There, Aβ directly attacks electron transport complex IV and key Krebs cycle enzymes and leads to the fragmentation of mitochondrial DNA (mtDNA). “Hijacking” of the electron transport brings about excessive formation of reactive oxygen species (ROS) and reactive nitrogen species (RNS). Their peroxidative attack on membrane lipids yields mitochondrial toxins (hydroxynonenal, isoprostanes, and malondialdehyde), which inhibits complex I. Furthermore, ROS and RNS are generated at complexes I/III. Mitochondrial membrane polarity (MPP) collapses, and permeability transition pores (Ψm) open. The leakage of compounds of prokaryote origins such as cytochrome-c and N-formyl-Methionine-Leucin-Phenylalanine (FMLP) activates caspases and initiates apoptosis. Mitochondrial toxins also inhibit GLUT1 and GLUT4 [17].

Together with energy deprivation and toxic derivative formation, excessive Ca^2+^ influx and hyperphosphorylation play equally important roles in this pathophysiological cascade [17]. Through Ca^2+^ entry and the formation of ROS, RNS, and lipid peroxidation products, Aβ stimulates kinases such as the p38 mitogen-activated protein kinase (MAPK), c-Jun N-terminal kinase (JNK), glycogen synthase kinase 3β (GSK-3β), and cyclin-dependent kinase 5 (CDK5). While JNKs and p38 MAPK cause apoptosis, the stimulation of GSK-3β and CDK5 together with its activator subunit p25 give rise to tau phosphorylation. Subsequently, these steps hinder tau protein assembly and cause hyperphosphorylated tau to detach. This leads to destabilization of microtubules and aggregation, disruption of axonal transport, and aggregation of hyperphosphorylated tau proteins into neurofibrillary tangles. Blockage of nutritional transport is accompanied by the failure of trophic signaling in the synaptic cleft since Aβ directly inhibits BDNF and NGF release. Then, microstructural changes follow, such as the reduction in the number of dendritic spines and the decrease in hippocampal synapses, which lead to mild cognitive impairment. Later, the progress of the disease brings about macrostructural changes due to widespread neuronal death. Shrinkage of several brain regions (most significantly the cognitive centers like the hippocampus) together with virtual enlargement of the ventricles is accompanied by a serious decline in cognitive functions. Ultimately, bed-ridden patients succumb to death caused by septic shock, which is brought about by aspiration pneumonia, decubitus, or ascending urogenital infections [17].

In summary, we would say that the sporadic forms of neurodegenerative conditions are not diseases, but inevitable consequences of the normal aging of the brain. In our view, the increase in incidences can be attributed to increased longevity and the enormous contemporary burden of environmental factors (stress, known or unknown forms of infections and intoxications, chronic diseases, psychiatric disorders, substance abuse, and sleep deprivation) [44,45,46]. However, a decrease in the formation and increase in the clearance of Aβ can significantly delay the progress of the condition. Since all the above-discussed environmental factors first cause an imbalance in the intricate equilibrium of neuropeptide control and the gut–brain axis [76,87,88], it is not a far-fetched idea that their modulation offers a viable opportunity in the palliative treatment of neurodegenerative “diseases”.

## 2. Neuropeptides, the Modulators of Different CNS Processes

Neuropeptide research started with the discovery of the effects of vasopressin [89] and oxytocin [90] and the identification of Substance P [91]. Since then, we have accumulated enough knowledge to separate them from the classical neurotransmitters [92,93,94,95]. Neuropeptides are obviously much larger molecules, and their metabolism is completely different from that of the small molecular transmitters. For instance, the available variety of ligands is larger, as numerous splice variants can derive from the original gene [93]. The spectrum of ligands is further diversified by the fact that neuropeptides are not taken back to the neuron by a specialized reuptake system. They are processed by peptidases, which yields further biologically active compounds [94]. This way, their half-life is more prolonged and especially their derivatives can diffuse far from their original release site. Therefore, they can act in a paracrine, an autocrine (i.e., post- and presynaptically), and an endocrine manner [94]. Due to the above-mentioned biochemical features, their effect develops slowly, but it is longer-lasting, which strongly resembles the activity of hormones [96,97]. That is why they are often referred to as neurohormones and their activity is described as “neuromodulation” [92,93,94]. Another specific feature of neuromodulation is that it is realized through a much broader array of receptors than that of the neurotransmitters. Moreover, the heterodimerization of their receptors can yield an infinite number of combinations and most versatile and flexible ways of signal transduction [93].

Thus, we can summarize that, from a functional point of view, neuromodulation strikingly differs from neurotransmission in the following aspects. 1. Neuromodulation is more flexible and resilient 2. It is not ephemeral, but much longer-lasting and often more profound (it does not have a reuptake system, and the breakdown often yields active metabolites) 3. It is more buffered, since one neuropeptide can easily take over the function of another one (due to this functional overlapping, knock-out animals do not show conspicuous symptoms). Moreover, it is very important to add that many peptide hormones, which are expressed in the CNS, have completely different activities than neuromodulators. For instance, ANP and BNP are potent regulators of salt–water homeostasis in the periphery, while in the CNS, they harness the stress axis [98,99]. On the other hand, vasopressin, the key regulator of fluid balance, has an opposite activity on the stress response [100]. Furthermore, while GHRH as a hormone indirectly and directly stimulates growth, as a neuromodulator, it is one of the most important regulators of the sleep–wake cycle [101].

The reaction of the CNS to challenges gives rise to a complete rewiring of the connectome [97,102]. An alteration of neuropeptide expression represents one of the earliest steps in the translational changes. The secretion of neurohormones is considerably altered by the steps of normal aging, such as menopause and senescence. Pathophysiologic alterations, which come with increased stress (processed or homeostatic challenges), will accelerate the aging of the brain [103]. Moreover, disorders of food intake, obesity, metabolic syndrome [104,105], and alterations of the gut–brain axis [87,88] have also been proposed as potential contributing factors. For instance, changes in corticotrophin-releasing factor (CRF), luteinizing hormone-releasing hormone (LHRH or GnRH), and growth hormone-releasing hormone (GHRH) secretion are apparently linked to neuroinflammation, obesity, and hyperinsulinemia. All these pathophysiological processes play a well-established role in the development of Alzheimer’s disease [106], though the details of causality are far from being clarified. Altogether, dozens of neuropeptides or neuropeptide families have already been implicated in the pathophysiology of Alzheimer’s disease [107]. However, in the present review, we will focus exclusively on those neuropeptides or neuropeptide families and their physiologic and pathophysiologic activities; these investigations conducted by the authors have significantly contributed to the literature. Therefore, the pathophysiologic role of orexins, neuromedins, RFamides, corticotrope-releasing hormone family, growth hormone-releasing hormone, gonadotropin-releasing hormone, ghrelin, apelin, and natriuretic peptides are discussed in detail.

## 3. GnRH

GnRH [108] and its receptor (GnRH-R) are not confined to the hypothalamic–pituitary axis [109]. In the periphery, the LHRH system coordinates gonadal functions and serves as a growth factor for many tumors [110,111]. In the CNS, a hypothalamic and an extra-hypothalamic cell population can be separated [112,113], where dense immunostaining for GnRH and GnRH-R can be demonstrated [112,113,114]. In addition to endocrine functions, these cells are involved in the regulation of the olfactory system, feeding, reproductive behavior, circadian rhythms [112,113,115], and the development of malignancies [116,117]. In vertebrates, both in physiologic and pathophysiologic circumstances, the actions of GnRH are mediated by three types of GnRH receptors, among which GnRH-I and GnRH-II are the mammalian variants. However, numerous splice variants can be detected in the population, some of which are associated with malignancies [118,119]. These receptors can activate either the Gα_q/11_ route or inhibit the cAMP-protein kinase A (PKA) signaling via Gα_i_ coupling.

The GnRH-positive subventricular zone [120] often shows not only steady hyperfunction [121] but also hypertrophy and hyperplasia, in the absence of steroid feedback, in postmenopausal and andropausal subjects [114]. Since the subventricular zone [120] is a frequent starting nidus of primary glioblastoma multiforme (GBM) and this age group has the highest prevalence of GBM [122], these data suggest a potential role of the GnRH system in CNS pathologies [116,117]. For instance, the majority of publications also suggest an initiating role of the GnRH–gonadotropin axis in postmenopausal cognitive decline [71,72,123]. It is well established that estrogen is neuroprotective, by maintaining neurons in the hippocampus [124,125], while elevated levels of gonadotropins and GnRH promote neurodegeneration [126]. Postmenopausal subjects, who suffer from a degeneration of LH receptor-expressing hippocampal neurons, shows an increase in plasma and CSF levels of luteinizing hormone (LH), follicle-stimulating hormone (FSH), and GnRH as compared to controls [127]. Regarding the pathogenesis of Alzheimer’s disease, the most notable signal transduction mechanism of the GnRH receptors appears to be the Gα_q/11_-elicited stimulation of PLC and Ca^2+^ entry as well as the activation of phosphotyrosine phosphatase (PTP) [119]. This ultimately may lead to Ca^2+^ toxicity, aberration of the cell cycle, activation of CKD5, increased formation of Aβ rafts, and hyperphosphorylation of tau protein [119,126,128]. The sequelae can be summarized as an increase in proliferation and failure of apoptosis of neoplastic cells and a concomitant decrease in autophagy and senescence of healthy neurons (Figure 2).

It is important to emphasize that not only the quantity of the hormone, but also the rhythm of secretion is crucial in proper hypothalamic regulation in both genders. GnRH is secreted in pulses, and higher frequencies favor LH, while lower frequencies support FSH release [129]. The pathophysiologic alteration of pulse frequency alone can give rise to common disorders such as polycystic ovary syndrome (PCOS) [130]. In the past few decades, studies have revealed that the regulation of pulsatile GnRH release is dependent on the activity of the intricate kisspeptin–dynorphin neuronal network, which is located in the anteroventral periventricular nucleus (AVPV) and the arcuate nucleus (ARC) [131]. While kisspeptin alone appears to act as a memory booster in the short term [132], in the long term, its failure appears to be responsible for the dysregulation of the GnRH–gonadotropin axis and steady augmentation of GnRH release. Perhaps the maintenance or correction of physiologic pulses would be even more beneficial in the prevention of neurodegeneration. Nonetheless, it is important to emphasize again that even male patients are not spared and those who receive androgen deprivation therapy are at risk of developing LOAD [133].

The modulation of the GnRH system is used for the treatment of several disorders, especially cancers of the gonadal system. GnRH super-agonists are the mainstay in the therapy of prostate cancer acting through the downregulation of GnRH-R [134]. Moreover, our previous studies showed that the GnRH antagonist Cetrorelix [135] and the cytotoxic analog AN-152 [136] can also be successfully used for the treatment of cancers of the reproductive system [136] and of other organs [137,138,139,140,141,142,143,144]. Considering the feasible harmful role of GnRH-R overactivation in the pathogenesis of Alzheimer’s disease, both animal experiments [145,146] and clinical trials [147,148] with GnRH analogs were carried out to check their effect on the progress of Alzheimer’s disease. Both forms of chemical castration (by GnRH super-agonists [147,148] and GnRH antagonists [145,146]) showed promising results.

Therefore, it appears that sexual steroid replacement therapy can be advantageous in both sexes and may delay the development of LOAD [149,150] since exogenous gonadal steroids suppress GnRH release. Another important aspect of this thread is that in the case of hormone deprivation therapy of gonadal cancers, in the long term, we should prefer chemical castration by GnRH agonist to treatment with antagonists of steroid hormones [145,146,147,148].

## 4. RFamides

The group of RFamide peptides encompasses neuropeptide FF (NPFF), neuropeptide AF (NPAF), neuropeptide SF [NPSF or gonadotropin-inhibitory hormone (GnIH)], prolactin-releasing peptide (PrRP), and the pyroglutamylated RFamide (QRFP) peptide [151,152]. Usually kisspeptin and its derivatives are also included in the family, although some recent publications dispute it [153]. The RFamide peptides all share an N-terminal sequence homology (Arg-Phe-NH_2_ motif at their C-terminus). Even though their distribution patterns are broad and somewhat overlap in the CNS, their difference in structure and receptor preference arms them with a unique activity spectrum. They bind to a whole family of receptors, with somewhat different affinities: NPSF/GnIH to the neuropeptide FF receptor 1 (NPFFR1 or GPR147), NPFF and NPAF to the neuropeptide FF receptor 2 (NPFFR2 or GPR74), QRFP to the pyroglutamylated RFamide peptide receptor (QRFPR or GPR103), kisspeptins to the kisspeptin receptor (KISSR1 or GPR54), and PrRP to the prolactin-releasing peptide receptor (PrRPR or GPR10) [151,152,154]. This way, despite some functional overlapping, their physiological activity spectra differ remarkably. For instance, while PrRP stimulates the hypothalamic–pituitary–adrenocortical (HPA) axis [6], stereotyped behavior [7], and pressor response [8], NPAF, in addition to inducing an HPA response and facilitating locomotor activity, decreases heart rate and core temperature [9]. These virtual discrepancies can be explained by the different receptor affinity of the peptides in this family. NPFFRs inhibit the adenyl cyclase–PKA signaling and stimulate the phospholipase C–inositol triphosphate–diacyl glycerolphosphate (PLC/IP3/DAG) route. On the other hand, the KISSR1, QRFPR, and PrRPR appear to facilitate both pathways. Ultimately, they can exert divergent actions on the MAPK-, JNK-, calcineurin-, and nuclear factor κB (NFκB)-regulated transmissions [155,156,157].

Thus far, kisspeptin [151,152] has attracted the most attention from the scientific community. This neurohormone [158] was isolated as the endogenous ligand of the orphan G protein-coupled receptor GPR54 (KISS1R) [159,160]. However, later, it was also verified to be able to bind to the NPFFR2 [161]. Kisspeptin mRNA is transcribed from the *kiss-1* gene, and the complete, translated peptide contains 54 amino acids (KP-54). The cleavage of the initial peptide gives rise to biologically active derivatives which consist of 14, 13, or 10 amino acids. This way, they were christened kisspeptin-14 (KP-14), kisspeptin-13 (KP-13), and kisspeptin-10 (KP-10) [158,159]. Although it was originally isolated from the human placenta, later studies demonstrated that kisspeptin and its receptors are widely distributed in the CNS. They are especially abundant in the limbic system, the striatum, and the hypothalamus, including the AVPV, the periventricular nucleus (PeN), the anterodorsal preoptic nucleus (ADP), the ARC, and the paraventricular nucleus (PVN) [159,160,162,163,164]. Kisspeptin and KISSR1 expression are also remarkably high in the pituitary gland. The inhibition of metastatic spread of malignant melanoma was the first physiological activity of kisspeptin [165]. However, since then, it has become obvious that its major activity is the regulation of the reproductive axis [131,166,167,168,169]. The kisspeptin, dynorphin-A, and neurokinin-B cosecretory (KNDy) neurons of the ARC and the kisspeptin-positive (Kiss1) neurons of the AVPV/PeN cooperate in the bimodal control of the GnRH neurons of the preoptic area (POA). Later studies also verified the activity of the cerebral and spinal kisspeptin networks in the control of the HPA axis, thermoregulation, pain sensation, stereotyped behavior, food intake, and learning [131,170,171,172,173,174].

The central role of the kisspeptin–dynorphin system in the control of the HPG (hypothalamic–pituitary–gonadal) axis [131,168] and the interaction between the kisspeptin neurons and other neuropeptide systems [175] suggest that RFamide peptides may also take part in the pathogenesis of neurodegenerative diseases. RFamide neurons together with the orexin network may accelerate [176,177,178,179], or in cooperation with ghrelin, decelerate [131,175,180,181] the progress of Alzheimer’s disease. Nonetheless, their direct effect appears unequivocally neuroprotective. Kisspeptin, PrRP, and QRFP appear to inhibit different steps of neurodegeneration. In vitro, it was demonstrated that kisspeptin release is stimulated by pretreatment with aggregation-prone proteins such as Aβ, prion protein, and amylin in neural cell cultures. Kisspeptin inhibited the neurotoxicity of Aβ, prion protein, and amylin. Further, the knockdown of the *kiss-1* gene enhanced the toxicity of the aggregation-prone peptides, while *kiss-1* overexpression was neuroprotective [182]. In vivo, KP-13 was able to reverse the memory impairment induced by Aβ [132]. Moreover, other RFamides also have beneficial effects. The long-term treatment of tau-overexpressing mice by the lipidized analog of PrRP significantly preserved spatial memory, neurogenesis, and synaptic plasticity. This lipoprotein also attenuated tau hyperphosphorylation, Aβ plaque load, cortical microgliosis, and astrocytosis in the hippocampus. These effects can be attributed to the tau-dephosphorylating activity of protein phosphatase 2A and the inhibition of caspase 3 [183,184]. Finally, the QRFP-orexin heterodimer receptor has proven to have strong neuroprotective activity via the activation of the extracellular signal-regulated kinase 1/2 (ERK1/2) pathway [185]. It is equally important to emphasize that acutely administered kisspeptin [132], NPAF [186], and QRFP [187] all enhance memory and facilitate learning in different experimental models. KP-13 facilitated memory formation and prolonged memory retention; NPAF improved the consolidation of passive avoidance learning [186], while QRFP improved short-term memory consolidation in the Morris water maze [187].

In conclusion, we can say that the RFamides seem to be promising targets of the pharmaceutical chemistry. Both their acute [182,183,185] and chronic [131,180,181] administration can be beneficial in the treatment of neurodegenerative diseases. However, a tremendous amount of work is required to separate the direct and indirect and acute and chronic actions of the various ligands on their receptor spectrum.

## 5. GHRH

The concept of neuroendocrine regulation and our knowledge about the activity of GHRH has changed dramatically over the past few years. The well-known hypothalamic trophormone GHRH has, apart from the stimulation of hypophyseal secretion, been described to exert various physiologic extrapituitary functions [188] promoting cell differentiation, wound healing, proliferation of immune cells, and sleep activation. It is very important to emphasize that these sorts of versatile physiologic activities are carried out in an autocrine, paracrine, and endocrine fashion and their receptors are located not only on GH-positive cells but also on IGF-secreting and target cells. Moreover, GHRH also plays an equally important role in pathophysiologic processes. Different forms of the GHRH receptor (GHRH-R), the pituitary type and the splice variant 1 (SV1), are expressed in several physiologic and pathologic cell types [189,190,191,192]. This way, GHRH can stimulate cell growth and cellular activity by direct action on its receptors or indirectly via the release of growth hormone (GH) and insulin-like growth factor-I (IGF-1) [188,189,190,191,192,193]. The direct activity is mediated by the formation of cyclic adenosine monophosphate (cAMP) via the coupling of the Gα_s_ subunit and activation of adenylyl cyclase. However, the coupling of Gα_q_ induces the release and entry of Ca^2+^ [194]. The indirect activity is dependent on the signal transduction of the GH receptor and the IGF receptor. The latter is a tyrosine kinase receptor, while the intracellular domain of the GH receptor binds Janus kinase (Jak) 2, which phosphorylates signal transducer and activator of transcription (STAT) proteins [195,196].

Recent experiments have shown GHRH to modulate cognitive processes in both physiologic and pathophysiologic circumstances; exogenous GHRH impaired hippocampal memory consolidation [197], while the inhibition of the GHRH-GH-IGF axis by GHRH antagonists showed a positive impact on learning and memory [145,146,198,199,200,201]. Though the literature reflects somewhat conflicting data regarding the etiopathogenetic role of the GHRH-GH-IGF system in the development of dementia [202,203,204,205,206], the intrinsic resistance to the action of the axis in Laron syndrome patients appears to confer a significant protection not only to cancers and diabetes but also to other detrimental consequences of aging [207,208]. We propose that the advantageous activity of GHRH antagonists can be attributed to their potential to inhibit the secretion of IGF, the oversecretion of which can be equally detrimental as the introductory hyperinsulinemic stage of type II DM [49,50]. This hypothesis was tested in multiple sets of experiments in which the top regulator of the GHRH-GH-IGF axis was neutralized by MIA-690, a powerful GHRH antagonist. The treatment successfully mitigated the progress of Alzheimer’s disease in FAD mice [50]. Functional memory tests using Morris water maze (MWM) [50] and morphological assays using immunohistochemistry [209] both verified our concept.

It appears that in insulin resistance, the derailed activation of IGF receptors by hyperinsulinemia will dysregulate two important biochemical pathways and can explain our findings [210,211,212]. The first is the pathophysiologic activation of the mammalian target of Rapamycin (mTOR)—Unc (uncoordinated)-51-like autophagy-activating kinase 1 (ULK1) pathway. The most important activity of the mTOR-ULK1 cascade is the inhibition of autophagy. The pathway is involved in normal cellular senescence, aging, and pathophysiologic conditions such as oxidative, inflammatory, degenerative, and neoplastic processes [17,192,201,203,204,206,213], including Alzheimer’s disease [210,214]. Second, stimulation of the IGF receptors inhibits the Ak strain transforming (Akt)-mediated inhibition of GSK-3β [215], which, in normal circumstances, is one of the most significant stimulators of autophagy [216]. Moreover, as it was detailed above, GSK-3β is the most important culprit in tau hyperphosphorylation [217]. Further, it cannot be emphasized enough that in Alzheimer’s disease, the disruption of beneficial autophagy is a key moment of the pathogenesis and metabolic derangement can be observed early in the course of the disease even in patients with mild cognitive impairment. This is because ROS generation of damaged mitochondria activates the phosphoinositide 3 kinase (PI3K)/Akt/mTOR pathway. This leads to the inhibition of the mitophagic removal of damaged mitochondria. Since the source of ROS is the mitochondria themselves, suspension of the mTOR-ULK-1 pathway fundamentally impedes the mitophagic self-healing of neurons [17,216]. Moreover, autophagic removal of misfolded intracellular rafts also appears to be a key component in the prevention of Alzheimer’s disease [218]. Therefore, in hyperinsulinemia in prediabetes, a reduction of insulin/IGF-I signaling can delay the onset of protein aggregation-mediated neurotoxicity [203,204,213,219].

## 6. Ghrelin

In the field of neuroendocrinology, the discovery of ghrelin has been one of the most important achievements of reverse pharmacology [220]. It has been identified as the endogenous ligand for a previously known, but orphan G protein-coupled receptor [220]. After translation, the peptide undergoes post-translational modification by ghrelin O-acyltransferase (GOAT), which yields the ultimate active form of acyl-ghrelin [221]. The actions of acyl-ghrelin are mediated by the ghrelin receptor or growth hormone secretagogue receptor (GHS-R), the splice variants of which (GSH-R1a and GSH-R1b) form heterodimers with other receptors such as the somatostatin receptor 5, the dopamine receptor type 2 (DRD2), the melanocortin-3 receptor (MC3R), and the serotonin receptor type 2C (5-HT2c) [222]. Nonetheless, even a single unit or a homodimer can stimulate multiple signal transduction pathways through either the G proteins Gα_q/11_, Gα_12/13_, and Gα_i/o_ or recruiting β-arrestin. Subsequently, the coupling of Gα_q/11_ gives rise to Ca^2+^ release and the activation of PKCβ, calcium calmodulin-dependent protein kinase IIα (CamKII), and 5′ adenosine monophosphate-activated protein kinase (AMPK). On the other hand, Gα_i/o_ coupling causes a stepwise induction of PKC then PKCε, PKA, and Akt, while recruiting β-arrestin leads to the stimulation of ERK and Akt [223]. However, the picture is more complex, since some studies demonstrated the effects of other splice variants (such as obestatin) of the ghrelin gene and the intrinsic activity of des-acyl ghrelin itself, whose actions are possibly mediated by the CRF receptors [224].

Initial experiments disclosed the role ghrelin plays in the regulation of feeding [225,226,227]. However, its widespread distribution in the CNS [228,229] clearly suggests that the peptide should serve as a much more general transmitter in neuroendocrine regulation. Especially the expressions of ghrelin and its receptor in the hypothalamus and the pituitary gland [230] pointed to a prominent role of this neuropeptide in the regulation of behavioral, endocrine, and homeostatic processes. Since then, several publications have revealed that ghrelin is an important regulator not only of feeding but also of hypothalamic–pituitary–target organ axes, thermoregulation, and behavioral processes [231,232,233,234]. It is of special importance that ghrelin has proven to improve memory in numerous experimental settings, and it has significant neuroprotective features [235]. Further, hippocampal ghrelin and GOAT expression [236] are significantly diminished in Alzheimer’s disease. Obviously, this may be a simple consequence of neurodegeneration. However, the finding that point mutations of the ghrelin gene were associated with the increased propensity to Alzheimer’s disease [237] raised the possibility that deficiency of the ghrelinergic tone can be a causative factor of neurodegeneration, especially in obese subjects who show a paradoxical resistance to ghrelin, which, coupled with leptin and insulin resistance, reflects a decreased resilience of hypothalamic body weight regulation [238]. This way, decreased ghrelin signaling can also be blamed for the development of obesity-related neuropsychiatric conditions such as anxiety disorders [239], schizophrenia [239], major depression [239,240], epilepsy [241], MS [242,243], ALS [244], Parkinson’s disease [245,246], Huntington’s disease [245,247], and Alzheimer’s disease [239,245,248]. Deprivation of ghrelin signaling can be harmful because ghrelin, like apelin (see later), is cytoprotective [249]. It triggers autophagy [250,251,252] and activates the ubiquitin–proteasome system (UPS) [250], which leads to the concomitant removal of misfolded Aβ [250]. It also stimulates neuronal glucose uptake [253] and accelerates synaptogenesis and neurogenesis [254,255]. These actions are mediated by the inhibition of the GSK-3β-catenin [252,253,256,257,258] pathway [259]. Additionally, ghrelin inhibits the Nod-like receptor protein 3 (NLRP3) inflammasome activity [260], apoptosis [251,261,262], Glu [263] and Ca^2+^ toxicity [264,265], tau hyperphosphorylation [257], superoxide anion production, and mitochondrial dysfunction [264,265]. Blockage of pathophysiologic NMDA signaling is of special importance since it protects against a decrease in BDNF secretion, which is very characteristic in major depression, Huntington’s disease, and Alzheimer’s disease [266,267,268,269].

Moreover, dysregulation of ghrelin homeostasis can also directly influence cholinergic neurotransmission. Upregulation of butyrylcholinesterase (BChE) in obesity and consequent ghrelin resistance has a two-pronged pathophysiologic effect. This is because BChE not only acts as the major breakdown enzyme of acetyl-ghrelin, the active form of the peptide [270], but also as an alternative acetylcholinesterase (AChE) [270,271]. Therefore, BChE hyperactivity can, at the same time, decrease the beneficial neuroprotective tone of ghrelin and block the essential cholinergic input to the hippocampus [78,79]. This way, while treatment with ghrelin, acylated ghrelin, or synthetic agonists [181,253,256,272] appears advantageous in the long run, the inhibition of BChE [271,273] may provide both long-term and short-term beneficial effects in Alzheimer’s disease. So far, animal experiments have returned promising data. The administration of a ghrelin agonist or acylated ghrelin decreased Alzheimer’s disease-related cognitive impairment [274], which may be attributed to its beneficial activity against the harmful effects of Aβ on the LTP [275] and the synaptic degeneration of cholinergic fibers [255].

## 7. CRF and Urocortins

The stress response or general adaptation syndrome (GAS) was described by Selye [276] in 1935. Since then, it has been acknowledged as one of the most general physiologic and pathophysiologic responses of living organisms. The GAS is controlled by one of the central neuroendocrine axes later described by Schally’s and Guillemin’s groups [277,278]. The input of the HPA system is diverse, and numerous neurotransmitters and neuropeptides take part in the signal transduction toward the hypothalamus [279]. Catecholamines, cytokines, NPY, neurotensin (NT), ghrelin, apelin, and endomorphins stimulate [234,279,280,281,282,283,284], while GABA, oxytocin, glucocorticoids, ACTH, endorphins, enkephalins, and natriuretic peptides inhibit the system [98,99,285,286,287]. This plethora of mediators provides a versatile and flexible means to translate the modality (systemic and neurogenic ones) and the schedule (acute, repeated, and chronic) of the challenges to the hypothalamus [98,99,288]. The systemic stressors (e.g., metabolic, osmotic, and immune) are directly projected to the brainstem, while the neurogenic paradigms (fear, anxiety, and pain) are processed by higher cerebral centers such as the components (amygdala, hippocampus, etc.) of the limbic system [288]. In sharp contrast with the aforementioned features, the output of the GAS is quite uniformly organized and can be simplified as a two-pronged response. The CRF-positive neurons of the PVN mediate the responses to the acute and processed stimuli, while the release of arginine vasopressin (AVP) from the parvocellular groups in the PVN and the supraoptic nucleus (SON) are responsible for managing the chronic, repeated, and homeostatic challenges [100]. AVP then activates different subtypes of the vasopressin receptor 1 (V1R), especially the vasopressin receptor 1B (V1BR), which is predominantly expressed in the pituitary gland [289,290].

A detailed experimental analysis of the CRF peptide family and the CRF receptors (CRFRs) has revealed that the central regulation of the CRF pathway is more complex. In humans, five further peptides of the family are released, which are called urotensin II (UTN II), urotensin-related peptide (URP), urocortin 1 (UCN1), urocortin 3 (or stress-coping, UCN3), and urocortin 2 (or stress-coping-related peptide, UCN2) [291,292]. The members of the family bind to the urotensin 2 receptor (UR-II-R), the 2 CRFRs (CRFR1 and CRFR2), and their splice variants with distinct preference [291,293]. The spectrum of the receptors is greatly expanded by numerous splice variants, among which soluble versions of the receptors can be detected in the circulation [291,292]. This makes the whole system extremely resilient. In general, we can say that CRF and UCN1 acting on the CRFR1 are the main stimulators of the HPA axis. It is also worth mentioning that one member, UTN II, can elicit a significant central release of CRF [291,292]. A modulatory built-in feedback mechanism of the system is mediated by CRFR2, which preferentially binds UCN2 and UCN3. According to literature data, the latter ligands appear to dampen the anxiogenic and GAS-inducing activity of the CRFR1-selective ligands [291,292]. That is why UCN3 is also known as a stress-coping and UCN2 as a stress-coping-related peptide, both of which appear to represent the first-line modulators of the HPA response. However, these antagonistic actions must be related to the distinct distribution pattern of the ligands and the receptors, because the signal transductions of the CRFR1 and the CRFR2 are practically the same [294], which mainly involves the coupling of Gα_s_ and the induction of the PKA–ERK–MAPK pathway. The CRFRs can also couple with Gα_q11_, Gα_i/o_, Gα_i1/2_, and Gα_i/z_. In general, these routes inhibit adenylyl cyclase and give rise to the activation of the PLC/DAG/IP3 pathway and the concomitant increase in cytosolic Ca^2+^ and the stimulation of ERK1/2 and PKC [294]. This above-described CNS mechanism is complemented by the ultrashort, short- and long-loop feedbacks of CRF, ACTH, and cortisol [288]. Therefore, the built-in dampening mechanism such as stress coping through UCN3, the plethora of feedback loops, and the intrinsic antiphlogistic activity of the glucocorticoids protects against the development of cytokine storms and the self-destructive, severe, inflammatory response (SIRS) [288,294,295].

The widespread expression of CRF and AVP receptors in the CNS, especially in the limbic system, also explains their robust behavioral activity. Their extremely important role in the regulation of emotions, affections, anxiety, etc., is undeniable [296]. The CRF family, due to its central role in anxiogenesis and the stress response, has been implicated in the pathogenesis of numerous neuropsychiatric disorders such as major depression, bipolar disease, chronic anxiety, panic disorder, and obsessive–compulsive disorder (OCD) and in the etiology of many neurodegenerative and neuroinflammatory diseases such as Alzheimer’s disease, Huntington’s disease, Parkinson’s disease, progressive supranuclear palsy (PSP), and MS [103,297,298]. The most important association has been known for decades: in major depression, chronic anxiety, bipolar disease, and psychotic episodes of major psychiatric disorders, the activity of the HPA system dramatically increases [299,300,301]. In addition to general overactivity, the disruption of the normal circadian pattern is the most conspicuous finding. Additionally, the HPA axis of depressed patients cannot be suppressed by such strong and long-acting synthetic glucocorticoids as dexamethasone [302]. This may reflect a central resistance to glucocorticoid feedback [303], mainly due to mutations of the glucocorticoid receptor, NR3C1 (nuclear receptor subfamily 3, group C, member 1), and one of its chaperone molecules the tacrolimus (FK506)-binding protein 51 (FKBP51) [304]. Therefore, it is not surprising that hormonal changes precede and may predict the changes in clinical status [305]. This way, the pharmacologically treated patients, who do not respond to a given drug, can be shifted to a different antidepressant regime without unnecessary delay. Owing to this temporal relationship, it is not surprising that the alterations of the HPA system are suspected to play a causative role in chronic neuropsychiatric diseases. In the literature, the CRF family, AVP, the glucocorticoids, and the glucocorticoid receptors have all been implicated in the pathogenesis of depression [303]. The flexibility and responsiveness of the system are dramatically changed by early-life stress or somatic disorders, and these alterations will be branded in the patients’ genome by epigenetic alterations [299,303,304,306]. In sharp contrast with CRFR1, which appears to mediate adverse effects in neurodegeneration, some lessons learned from the pathogenesis of Parkinson’s disease and ALS suggest that CRFR2 activation may confer protection against neuronal apoptosis [307,308]. It is especially important, owing to the fact that the hippocampus and the limbic system in general boast one of the densest expressions of CRFR2 in the brain [294,296].

Regarding Alzheimer’s disease, even the earliest studies unveiled that CRF expression and interstitial CRF levels are sharply decreased in the cortex of the patients, who suffered from an advanced stage of the disease [309,310,311], which is not surprising, taking the massive loss of neurons in the mesolimbicocortical structures into consideration. However, later publications demonstrated that in the early stages, the opposite alterations can be observed. Since then, the pathogenetic role of HPA axis overactivation and excessive CRF release have been unequivocally accepted, though the exact pathomechanism is far from being clarified. Some researchers argue that glucocorticoids themselves play the most important pathogenetic role [312,313,314], while others report equivocal data [297,303]. However, recent findings unambiguously support the view that the central overexpression of CRF and/or ACTH is the culprit, which directly and/or indirectly (via glucocorticoid hypersecretion) brings about detrimental consequences [315,316]. It is well known that glucocorticoids inhibit cell renewal and proliferation owing to their catabolic and apoptosis-inducing properties [317] and they can inhibit the normal microglial elimination of Aβ debris [318]. On the other hand, recent studies suggest that stress caused a decrease in synaptic plasticity and the neurodegeneration in the mesolimbic structures can be attributed mostly to the hyperactivity of the CRF neurons [319,320,321]. For instance, the triple transgenic mouse model that overexpresses both CRF and APP showed a significant increase in Aβ plaques in the cortex and hippocampus, as compared to APP mice. Moreover, the triple transgenic strain displayed substantial decreases in dendritic branching and dendritic spine density in pyramidal neurons. This was confirmed by behavioral studies in which the triple transgenic mice showed significantly impaired working memory and contextual memory [322]. Another study similarly assigned a causative role of the CRF system to Alzheimer’s disease. A double-transgenic mouse strain was crossed with CRFR1 null mice. Knocking out CRFR1 significantly reduced the Aβ burden in the hippocampus and insular, rhinal, and retrosplenial cortices [323]. Moreover, in a different study, CRFR1 antagonist treatment mitigated Aβ deposition, impaired synapses, and delayed cognitive deficits in a mouse model of Alzheimer’s disease [324]. In vivo studies also revealed that isolation and restraint stress increase Aβ levels in a CRF-dependent manner [325]. It appears that excessive CRF release in chronic stress gives rise to DNA damage in the CNS. Then, the accumulation of DNA damage drives the upregulation of cell cycle checkpoint protein kinase 1 (CHK1) and of cancerous inhibitor of protein phosphatase 2 (CIP2A), leading to Aβ raft formation [326]. However, other researchers suggest that restraint, isolation, and swimming stress accelerate neurodegeneration via mainly tau hyperphosphorylation [327,328]. Moreover, mice exposed to chronic noise showed CRF and CRFR1 overexpression, which caused GSK-3β activation and tau hyperphosphorylation [329,330].

## 8. Hypocretins/Orexins

The hypocretins/orexin system represents a complex and somewhat unique neuropeptide network in the CNS [331,332]. The cell bodies of the system are confined to a few specific regions of the hypothalamus (the lateral, the dorsal, the dorsomedial, and the perifornical areas). In sharp contrast, its receptors are scattered throughout the whole CNS and its axon terminals reach distant regions [333,334,335].

The family consists of two ligands (orexin-A and orexin-B), which belong to the structurally diverse incretin family [334,336]. Even orexin-A and orexin-B show only 50% structural overlapping. They bind to two receptors (OX1R and OX2R) with significantly different receptor affinities [334,336,337]. This can be attributed to the fact that the orexin receptors (OXRs) are identical only in 64% of their primary structure [338,339]. At first, the pharmacodynamic properties of orexin-A seemed more promising. It has a longer half-life because it comprises an N-terminal pyroglutamate residue and two disulfide bonds, which make it more resistant to proteolytic degradation. Orexin-A, since it is more hydrophobic, can pass the BBB more effectively [340]. OX1R and OX2R share their signal transduction mechanisms so the difference between their distribution pattern appears to be responsible for their unique action. They are G protein-coupled receptors (GPCRs), and their activation in the hypothalamus, via coupling to the Gα_s_, Gα_q_, and Gα_i_ family proteins, causes both cAMP and DAG-IP3 generation. Nonetheless, it is worth mentioning that the orexin receptors can cause Na^+^ and Ca^2+^ influxes via direct stimulation of ion channels such as the non-selective cation channels (NSCCs) or the Na^+^/Ca^2+^ exchanger (NCX) [341].

The first physiologic functions associated with orexins were the stimulation of hedonic feeding [336,342] and arousal [334,343]. One of the most important outputs of the orexin-positive cells targets several important centers of the ascending reticular activation system (ARAS), such as the lateral dorsal tegmental (LDT) and the pedunculopontine tegmental (PPT) nuclei in the mesopontine tegmentum (MT) as well as the locus coeruleus (LC), the nucleus raphe (NR), and the periaqueductal grey (PG) [335,344]. Nevertheless, the discovery was a milestone in neuroscience that narcolepsy and cataplexy observed in dogs [345], mice [346], and even humans [347,348,349] can be caused by either congenital [345,346,350] or acquired defects [351] of the orexin-hypocretin system. The latter point is especially important because it draws attention to the potential autoimmune impairment of the system caused by infections or vaccination against influenza or COVID-19 [351,352]. Further studies verified that numerous arousal-related functions are under the circadian control of the orexin system. In addition to food intake, the most important ones are fluid intake [353,354,355], metabolism, thermoregulation [356], activity of the HPA axis [357,358,359,360,361,362,363], reproduction [364,365], and threat-related adaptive behavioral processes [366,367,368,369,370,371,372]. Since sleep disorders, obesity, type II DM, permanent anxiety, chronic stress, and PTSD are all implicated in the pathogenesis of Alzheimer’s disease [73,74,75,76,373,374], it is not surprising that numerous experiments have been conducted to clarify the putative role of orexins in the development of Alzheimer’s disease [375,376]. Acutely, orexins are apparent memory boosters [369,370,371,372,377] and the integrity of the orexin network is required for proper memory consolidation [378]. However, chronic malfunction of the orexin network in Alzheimer’s disease goes with Janus-faced consequences. It appears that the initial stage of Alzheimer’s disease is characterized by a transient “flare” of the dying network. In this phase, increased CSF concentrations of orexins can be measured, which are presumably released from the injured neurons [379,380]. This is symptomatically reflected by agitation, restlessness, and a disturbed sleep–wake cycle. It is especially important to emphasize that sleep/wakefulness problems usually appear well before cognitive decline and sleep deprivation alone is a well-known accelerating factor in the progress of Alzheimer’s disease [44,45,374]. In our opinion, decreased glymphatic flow [44,45], alteration of gut microbiota [381], and direct pathobiochemical consequences of orexinergic hyperstimulation synergistically decrease the elimination of Aβ and augment the formation of neurofibrillary tangles [261,374,382]. Indeed, augmented orexinergic tone was demonstrated to inhibit microglial phagocytosis and the consequent Aβ degradation [176]. Then, increased membrane permeability and decreased reuptake of excitatory amino acids lead to glutamate and Ca^2+^ toxicity, hyperphosphorylation of tau protein, and apoptosis [17,374,382,383,384]. Another important result of recent studies is that the orexins can activate both the BACE1 and BACE2 enzymes [177,178]; therefore, they can also stimulate Aβ formation. This way, at the early stage of the disease, it appears that the treatment with an orexin antagonist may offer not only palliative (restoration of the sleep–wake cycle) but also causative (decrease in Aβ deposition) opportunities [338,373,385].

However, as mentioned above, the normal orexinergic tone is definitely neuroprotective [261,386] and increases hippocampal memory formation confirmed by numerous experiments [369,370,371,372,377,378]. The neuroprotective activity may be exerted through RF-amide receptors, to which the orexins show the highest structural similarity. The NPFF receptors belonging to the RF-amide peptide family, showing approximately 30% homology to OX1R and OX2R [34,35]. The heterodimerization of the orexin and the pyroglutamylated RF-amide receptors (GPR103) apparently confer the highest neuroprotective activity [185], through the stimulation of ERK/MAPK, NF-κB, PI3K-Akt, and Jak-STAT [185]. This way, at later stages of the disease, the lack of the protective tone in the exhausted and then spent network may have equally detrimental effects on neuronal viability as the original overexcitation during “flare”. Nonetheless, as some of the presented studies have demonstrated contradicting results on the role of orexins in Alzheimer’s disease, further experiments are warranted to clarify the role of orexins in different stages of the pathogenesis and to firmly establish the therapeutic opportunities offered by orexin analogs. Moreover, the dysfunction of the orexin system was demonstrated not only in Alzheimer’s disease but also in chronic neuropsychiatric conditions such as schizophrenia, bipolar disease, epilepsy, and drug addiction; in neuroinflammatory diseases such as MS, neuromyelitis optica, and autoimmune narcolepsy; and in numerous neurodegenerative disorders in addition to Alzheimer’s disease, such as dementia with Lewy bodies, Parkinson’s disease, FTD-ALS spectrum, and Huntington’s disease [375,387,388,389,390,391,392,393]. The orexin system, which consists of only a limited number of units restricted to relatively small regions, appears to be specifically vulnerable to neurodegeneration. Several morphological studies have verified the conspicuous loss of orexinergic neurons in these chronic CNS conditions [392,394]. That is why some shared symptoms, which reflect the abnormality of arousal, may result from the progressive failure of the orexin network. Changes in sleep patterns and vigilance are commonly found symptoms in MS and Alzheimer’s disease. Dysfunction of postural reflexes and cataplexy are almost universal findings in Parkinson’s and Huntington’s diseases. Rapid mood fluctuations, unaccountable anxiety, irrational fears, agitation, and extreme irritability are the common behavioral symptoms [395,396] in all neurodegenerative diseases. Later studies connected either the initial overexcitation or the spent-out phase of the degeneration of the orexin system to the above-described protean symptoms [387,390].

## 9. Apelins

Apelin [397] was identified as the first endogenous ligand for the so-far orphan G protein-coupled receptor, APJ [398]. Since then, the receptor has proven to bind apelin-36, apelin fragments, and another neurohormone, Elabela [399]. Apelin-36 was isolated from bovine stomach extracts and is processed from a 77 amino-acid precursor, preproapelin, which is cleft into several molecular forms in different tissues. Synthetic C-terminal fragments of preproapelin consisting of 13–19 amino acids were found to exhibit significantly higher activity at the receptor than that of apelin-36 [397,400]. The expression of the rat apelin receptor has been detected throughout the CNS [401]. Especially high concentrations were found in the hypothalamus, the pituitary gland, and the limbic structures. In the hypothalamus, the most intense expression is in the supraoptic and paraventricular nuclei [402,403], which suggests a prominent role of apelin in the regulation of homeostatic, behavioral, and endocrine processes. The signal transduction of APJ is quite diverse due to its several splice variants and its propensity to form heterodimers especially with the kappa opioid receptor (KOR). Coupling to Gai/o inhibits the adenylyl cyclase and, in turn, PKA. Parallel activation of phosphoinositide 3-kinase (PI3K) results in the stimulation of Akt and endothelial nitric oxide synthase (eNOS) yielding nitric oxide (NO). Ultimately, coupling to Gα_q/11_ activates phospholipase C (PLC) leading to the stimulation of the IP3/DAG/PKC and the ERK1/2 routes, whose pathways, in turn, increase the cytosolic Ca^2+^ level. Different splice variants, various G protein couplings, and the heterodimerization of APJ with KOR can confer tissue-specific action to the action of different apelin isoforms (apelin-12, apelin-13, apelin-17, and apelin-36) [404].

First, the effects of apelin on the circulation and the heart were described: the administration of apelin-13 increased water intake [405,406], and it was demonstrated to have an impact on blood pressure [405,407] and vasopressin release [406,408]. Then, other studies verified their role in the control of appetite, thermoregulation, pituitary hormone release, and behavior [283,406]. Regarding Alzheimer’s disease, perhaps the cytoprotective activity and memory-enhancing effects of apelins are the most promising facets of their functional spectrum [219,409,410]. They prevent the harmful consequences of oxidative stress [219,411,412,413], exposition to cytotoxic agents [219,413], and inflammation [414,415]. These activities of apelin are mediated by the pro-survival signaling via IP3, PKC, and mitogen-activated protein kinase 1/2 (MAP1/2) [416]. Apelin also inhibits excitotoxicity by attenuating NMDA receptor signaling [417]. Therefore, it is not surprising that in the past few years, several studies have been conducted to examine its putative, beneficial activity in neurodegenerative diseases [418]. In general, apelin has proven to have strong neuroprotective activity [418]. It promotes autophagy [219,413,419] while inhibiting apoptosis [219,412,413] and necroptosis [420]. Moreover, it stimulates the release of a well-known neurotrophic factor, BDNF [421,422]. Its beneficial effects have been demonstrated in several forms of chronic neuropsychiatric conditions, such as epilepsy [423], schizophrenia [424], and depression [425,426]; neuroinflammatory diseases such as MS [427]; and neurodegenerative disorders such as Parkinson’s disease [428,429], Huntington’s disease [430], Alzheimer’s disease [431], and ALS [432]. In Alzheimer’s disease, probably the most important component of its beneficial activity is the stimulation of mitophagic removal of the damaged mitochondria and autophagic removal of Aβ [433]. Therefore, apelin derivatives offer a viable therapeutic option for the treatment of these chronic progressive neurological conditions [418,425,433,434,435] such as Parkinson’s disease [436,437], Huntington’s disease [430], Alzheimer’s disease [414], and ALS.

## 10. Neuromedins

Neuromedins represent a versatile group of neuropeptides. They belong to the tachykinin family, and nowadays, some members are classified as brain–gut neuropeptides [438]. Since they are expressed in different organs and target different classes and subclasses of receptors, their physiological activity spectrum differs remarkably [439]. The first members were isolated from porcine CNS [440,441]. Since then, they have been classified according to their similarities to other tachykinins in structure and receptor preference (K—kassinine-like, B—bombesin-like, N—neurotensin-like, etc.). The neuromedins are abundantly expressed in those CNS regions (the hypothalamus and the limbic system) [442,443,444], which control the endocrine, the autonomic, and the behavioral processes. In that sense, neuromedin-B is the poster child of the group [445,446]. Neuromedin-N (NMN) seems to have a more specific activity spectrum. It was demonstrated to stimulate the HPA axis [447,448] and also displayed marked action on thermoregulation [449]. The first physiologic activity associated with neuromedin-U (NMU, U refers to the womb) was the potent, smooth, muscle-stimulating effect exerted on uterine tissue [442]. However, in complete agreement with its profound neural expression [450], NMU was verified to play a significant role in the regulation of core temperature, feeding [451], circulation [452], HPA axis [448,453], and behavior. Moreover, it increases overall locomotor activity, decreases sleeping [451,453,454,455], facilitates learning [456], and stimulates anxiety-related stereotyped behavior [453,454,455,457]. Neuromedin S (NMS) is a recent addition to this peptide family [458]. Its designation derives from its localization, since the distribution of NMS is confined to the suprachiasmatic nucleus (SCN) of the hypothalamus. NMS shows the highest structural similarity to NMU, but they are coded by two different genes [458]. As a conclusion, we can state that from a neurobiological point of view, NMU and NMS are the most important members of the group. Both peptides bind to the NMU-1R, which may explain the overlap in their functional spectrum, but NMS shows a specifically high affinity to the NMU-2R [458]. Therefore, NMU-2R appears to transmit the genuine activities of NMS [459,460]. Upon stimulation by both NMU-1R and NMU-2R, there is predominantly coupled Gα_q/11_ with some evidence of Gα_i_ coupling, which suggest that their primary action is the parallel inhibition of the cAMP–adenylyl–cyclase–PKA pathway and the stimulation of the PLC-IP3/DAG-Ca^2+^-PKC signaling route [461]. The expression of the NMU-2R is the highest in the hypothalamus [462,463], the thalamus, the cerebellum, the amygdala, and the hippocampus [464]. This pattern suggested a physiologic activity spectrum, which was later confirmed by functional studies. NMS was demonstrated to influence the circadian rhythm [458,465], feeding [466,467], hypothalamic hormone (e.g., GnRH and CRF) secretion, circadian rhythm of temperature, and behavior [282,468].

Since the arousal-controlling orexin network receives its most important input from the SCN [459,469,470], it is not far-fetched to suggest that the neuromedins may relay those photic stimulations to the orexin neuron, which arrive at the SCN via the retinohypothalamic pathway [458,471]. The significance of the orexin system, outlined in the previous chapter, in the pathophysiology of Alzheimer’s disease and the disintegration of SCN activity in neurodegenerative diseases [472,473], suggest a crucial role of the SCN and NMS in the pathogenesis of dementias. Presumably impaired circadian rhythm generated by a malfunctioning SCN might lead to an abnormal fluctuation of mood and sleep, which could impair glymphatic flow and initiate neurodegeneration [474]. Although very little data are available regarding the involvement of neuromedins in the development of Alzheimer’s disease, it is noteworthy that mild cognitive impairment can be associated with increased levels of neuromedin activity [475], despite the fact that neuromedins themselves appear to be intrinsically neuroprotective [476]. This activity spectrum is strikingly like that of orexins and confirms our hypothesis that the neuromedin–orexin networks cooperate in their cognitive actions similarly to their feeding- [471] and arousal [477]—related functions. Therefore, sleep disturbances appear not only secondary [44] but also primary, initiating factors in the pathogenesis of neurodegenerative diseases. Nevertheless, due to scarcely available and contradictory data [478], this aspect of tachykinin pathophysiology must be further scrutinized experimentally.

## 11. Future Therapeutic Perspectives

In recent decades, experimental findings have significantly transformed our understanding of Alzheimer’s disease. As discussed in the first chapter, our group views the disorder as a widespread, multi-etiologic process, with numerous individuals carrying polygenic factors with incomplete penetrance. This conceptual shift has forced some researchers to rework their experimental strategies, which earlier focused on direct interventions targeting the production of Aβ and neurofibrillary tangles. In our opinion, indirect approaches, which delay the inevitable deposition of β-pleats and preserve the efficiency of physiologic removal, should have greater therapeutic potential. That is why the neuropeptide segment of Alzheimer’s research has undergone dramatic changes. In the past few decades, several neuropeptides and synthetic neuropeptide ligands have been proposed to possess promising physiological and pharmaceutical actions [107,261]. Over time, however, in most of the cases, their beneficial actions proved to be non-specific. It appears that their effects can be attributed to non-specific neuroprotective or non-causal palliative memory-enhancing effects. This goes for the promising actions of pituitary adenylate cyclase-activating peptide (PACAP), neuropeptide Y (NPY), galanin, thyrotropin-releasing hormone (TRH), calcitonin gene-related peptide (CGRP), and neurotensin [107,261]. Therefore, it appears that a dramatic conceptual revolution is required to harmonize the often conflicting or dismissible data. To our mind, first, the harmful endogenous and exogenous challenges, which initiate the neurodegenerative cascade, must be analyzed. If one or more neuropeptides seem to play a disproportionately significant pathogenetic role in Alzheimer’s disease, they could be potential targets of pharmaceutical studies and could produce effective modulatory compounds.

Despite these contradictions, in the last few decades of Alzheimer’s research, three firm pillars of neuropeptide pathophysiology have remained intact and have even gained more attention. First, the overactivation of the GnRH system must be mentioned. It affects almost the whole population due either to meno- and andropausal changes or to the treatment of steroid receptor-positive tumors. It seems that in meno- and andropause, careful sex steroid replacement therapy is beneficial even from a cognitive point of view, which can be explained by the direct neuroprotective activity of sex steroids and the concomitant inhibition of GnRH release [149,150]. However, direct suppression of meno-, postmeno-, and andropausal GnRH surges with pertinent GnRH analogs represent another treatment option [138,139,140,141]. Further, in the case of gonadal tumors, both forms of chemical castration (either with GnRH antagonists or with GnRH super-agonist) are better alternatives than gonadal steroid antagonists when taking the neuropsychiatric side effects into account [145,146]. Additionally, fine-tuning the system in the ill and the elderly with RFamide analogs may provide more effective treatment with fewer side effects.

The second pillar is represented by the way in which chronic and early-life stress makes the human brain more susceptible to chronic neuropsychiatric and neurodegenerative conditions [103,297,298]. By now, several experiments have verified the direct pathogenic role of the CRF-CRFR system and the complementary adverse function of ACTH and glucocorticoids in the development of depression and neurodegeneration [303]. Therefore, those neuropeptides, which can harness and modulate the HPA system, such as the natriuretic peptides, oxytocin [288], or synthetic ligands may also confer beneficial effects in the case of Alzheimer’s disease. As far as pharmaceutical derivatives are concerned, the most promising molecules seem to be the CRFR1 antagonists [303,324] and CRF-binding protein (CRFBP) analogs [479], the administration of which must be commenced well before the development of irreversible morphological alterations. In the future, the introduction of orally available CRFR1 or biologically engineered CRFBP could open future therapeutic windows, if BBB-permeable peptide or non-peptide ligands are developed. It is also worth mentioning that the HPA system-activating nature of some cytoprotective neuropeptides (PACAP, neurotensin, etc.) [480,481] seem to hinder their application, despite their verified NGF and BDNF release-stimulating activity [261].

The third pillar is the regulation of sleep, which through the discovery of glymphatic circulation and progress in orexin pathophysiology is receiving more and more attention. The orexins and some of those mediators (MCH, melatonin, and NMS) [482,483], which cooperate in arousal and circadian rhythms, have been reported as promising targets for modifying the pathophysiologic processes in Alzheimer’s disease. Especially the restoration of the function of the orexin network and sleep hygiene appear as effective alternatives. For that, the most suitable compounds are the orexin antagonist [483,484], because, like melatonin [485], both the natural ligands and the pharmacological analogs can freely pass the BBB [96,486]. That is why some orexin antagonists have already been approved by the Food and Drug Administration (FDA) as insomnia medications [487].

However, since the discovery of the brain–gut axis [488], the chapters of neuropeptide research have been completely rewritten and a fourth pillar has been added to the structure. Modification of the enteric microbiota gives rise to severe neuropsychiatric consequences [489]. These changes are translated and transmitted to the nervous system by the alteration of the spectrum of the brain–gut neurotransmitters and neuropeptides. At present, the most important participants are leptin, ghrelin, the GHRH-GH-IGF axis, incretins (such as glucagon-like peptide 1 (GLP-1), glucose-dependent insulinotropic polypeptide (GIP), cholecystokinin (CCK), and NPY. The system is deeply altered in conditions such as polygenic obesities, monogenic obesities (Prader–Willie and Kleine–Levin syndromes), metabolic syndrome, type II DM, and PCOS [490]. The gastrointestinal modification of the neuromodulatory tone to the CNS has already been implicated in several neuropsychiatric conditions [489]. Recalling the old proverb “You are what you eat” and paraphrasing Descartes’ aphorism “Sum ergo cogito”, we should arrive at the conclusion that our nutritional habits determine our way of thinking. Recent modification and fine-tuning of the brain–gut–enteric microbiota system have yielded some astonishing results, especially with the incretin analogs [491]. They could be beneficial by suppressing the progress of an insidiously developing metabolic or neuropsychiatric condition. For instance, GLP-1 agonists in addition to supporting weight and glycemic control, appear to have a direct positive impact on the patients’ circulation and CNS function [491]. Moreover, restoration of the gut microbiota and the function of the ghrelin-GHRH-GH-IGF axis may help prevent several chronic neuropsychiatric conditions [381,489].

In summary, we would say that to covet a single panacea, the “philosopher’s stone”, in order to cure Alzheimer’s disease is a rather futile effort. In our opinion, Alzheimer’s disease is only the tip of the iceberg, a civilization phenomenon, which has become so apparent in the past few decades, because environmental stress has increased and a longer lifespan has opened a wider window to its detrimental consequences. Nevertheless, already approved drugs, such as GLP-1 agonists, [492], orexin antagonists [493], and modulators of the ghrelin system [494,495] can already be used for palliative or preventive purposes in the treatment of Alzheimer’s disease. The ghrelin and orexin analogs are of special merit since they pass the BBB [96,486,496]. Moreover, in our opinion, the restoration of normal sleep hygiene, due to its immense positive impact on glymphatic circulation, is one of the most important and viable options in the prevention of Alzheimer’s disease. Furthermore, the orexin antagonists do not have the severe detrimental cognitive side effects of the traditional soporific medications such as barbiturates and benzodiazepines. In addition to their acknowledged hypnogenic activity, the orexin analogs are currently under investigation for the treatment of anxiety, panic disorder, major depressive, and binge eating disorder [487].

Modification of the availability of IDE by orally available GHRH, ghrelin analogs, and BChE antagonists could open future therapeutic windows. Nonetheless, as physicians, we should be very careful with some treatment regimens, which potentially tamper with the neuropeptide metabolism, because several of them may interfere with the decomposition of Aβ. For instance, both ACE and neprilysin take part in the breakdown of Aβ. Therefore, their combined blockade by vasopeptidase inhibitors in order to increase the level of vasodilators (kinins), decreases the formation of vasoconstrictors (angiotensin), facilitates natriuresis (inhibiting aldosterone production and increasing NP level), and can lead to unpredictable side effects [70,497]. Nonetheless, the picture is more complex since angiotensin and CNP appear to stimulate [287,498], while ANP and BNP seem to inhibit [99,498] the release of CRF, one of the peptidergic culprits in the pathogenesis of Alzheimer’s disease. Another example of a potential negative side effect is the direct and long-term inhibition of MMPs, while pharmacological stimulation of plasminogen can advantageously accelerate Aβ breakdown. Obviously, the latter pathway cannot be utilized in the long run. The potential sites of beneficial therapeutic interventions and adverse side effects of present and future pharmacological ligands are summarized below in Table 1.

## Figures and Tables

**Figure 1 ijms-25-13086-f001:**
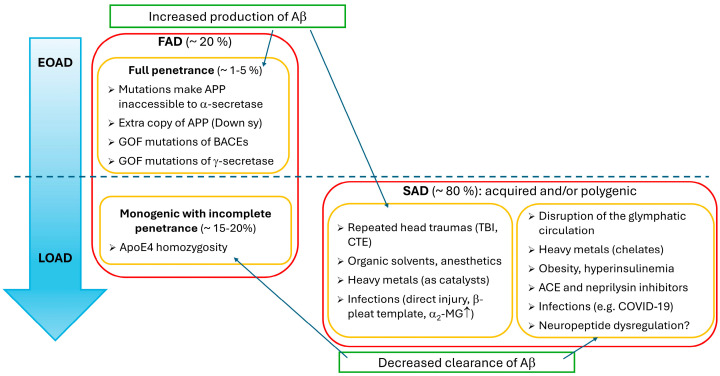
The outline of the pathophysiology of Alzheimer’s disease. APP: amyloid precursor protein; Aβ: amyloid-β; α2-MG ↑: increase in α2-macroglobulin; BACE: β-site APP-cleaving enzyme (β-secretase); COVID-19: coronavirus disease; Down sy: Down syndrome; GOF: gain of function; EOAD: early-onset Alzheimer’s disease; LOAD: late-onset Alzheimer’s disease; FAD: familial Alzheimer’s disease; SAD: sporadic Alzheimer’s disease; CTE: chronic traumatic encephalopathy; TBI: traumatic brain injury.

**Figure 2 ijms-25-13086-f002:**
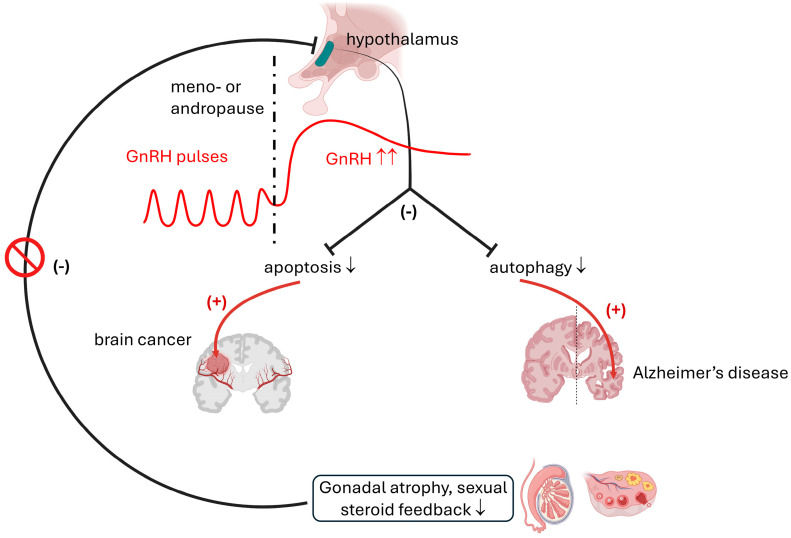
Schematic representation of the putative pathophysiologic consequences of GnRH overdrive in postmenopausal subjects. GnRH: gonadotropin-releasing hormone.

**Table 1 ijms-25-13086-t001:** Beneficial and adverse modulations of neuropeptide activities in Alzheimer’s disease.

Pathway	Target	Ligand or Substrate	Drug with Reference *	Potential Effect
HPA	CRFR1	CRFR1 antagonists	-	+
GHRH-GH-IGF1	GHRH-R	GHRH-R antagnists	-	+
HPG	GnRH-R	GnRH-R antagonists or super-agonists	-	+
insulin metabolism	IDE	insulin	NA	NA
RAAS	ACE	ACE inhibitors	Captopril, etc. [499]	- ?
NP metabolism	NEP	NEP inhibitors	Sacubitril [500]	- ?
matrix metabolism	MMPs	MMP inhibitors	Periostat [57]	- ?
GOAT-ghrelin	ghrelin-R	ghrelin-R agonist	Macimorelin [494]	+
	BChE	non-selective BChE inhibitors	Rivastigmine, Huperzine-A [495]	+
orexin	OX1R, OX2R	dual OXR antagnists	Suvorexant, Daridorexant, Lemborexant [501]	+

* FDA-approved drugs capable of passing the blood–brain barrier. ?: potential side effect in the long run; ACE: angiotensin convertase enzyme; BChE: butyrylcholinesterase; CRFR1: corticotrope-releasing factor receptor 1; GHRH: growth hormone-releasing hormone; GHRH-R: growth hormone-releasing hormone receptor; ghrelin-R: ghrelin receptor; GnRH: gonadotropin-releasing hormone; GnRH-R: gonadotropin-releasing hormone receptor; GOAT: ghrelin O-acyltransferase; IDE: insulin-degrading enzyme; NEP: neprilysin; NP: natriuretic peptides; OXR: orexin receptor; OX1R: orexin 1 receptor; OX2R: orexin 2 receptor; RAAS: renin angiotensin aldosterone system.

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
