# Peer review of "The Aggravating Role of Failing Neuropeptide Networks in the Development of Sporadic Alzheimer’s Disease"

_ijms, 2024, doi:10.3390/ijms252313086_

Round 1

Reviewer 1 Report

Comments and Suggestions for Authors

The author's review of Neuropeptides in sporadic AD is interesting; however, the manuscript needs improvements.

1. Manuscripts look mostly like a book chapter, not a review article.

2. The author descriptively provides all findings of peptides.

3. The author should have listed the findings of peptides in the table and focused more on molecular mechanisms and therapeutic approaches, which would have helped new researchers investigate.

4. It is better to provide figures for the mechanisms as well as a graphical abstract to grab the reader's attention.

5. The length of the review is too long for paying attention to information, it is better to cut it down to. 

Author Response

The author's review of Neuropeptides in sporadic AD is interesting; however, the manuscript needs improvements.

  1. Manuscripts look mostly like a book chapter, not a review article.

Thank you for the reviewers comments. We eliminated some repeats in the text and removed some overlapping references

2. The author descriptively provides all findings of peptides.

3. The author should have listed the findings of peptides in the table and focused more on molecular mechanisms and therapeutic approaches, which would have helped new researchers investigate.

In complete agreement with the reviewer, with all neuropeptides we added those signal transduction mechanisms, which could be responsible for beneficial or adverse effects of the peptides. They are also discussed in those sense, which biochemical points can serve as targets for pharmaceutical intervention 

4. It is better to provide figures for the mechanisms as well as a graphical abstract to grab the reader's attention.

In full agreement with the reviewer we made a graphical abstract, which summarizes the peptidergic mechanisms in the pahogenesis of Alzheimer's and incorporates the sites of putative therapeutic interventions, too.

5. The length of the review is too long for paying attention to information, it is better to cut it down to. 

We eliminated some repeats in the text and removed some overlapping references.

Reviewer 2 Report

Comments and Suggestions for Authors

Journal: IJMS (ISSN 1422-0067)

Manuscript ID: ijms-3281949

Type: Review

The manuscript title “The Aggravating Role of Failing Neuropeptide Networks in the Development of Sporadic Alzheimer's Disease” by Miklós Jászberényi et all, stands out for its clarity, breadth of study, and potential to considerably advance our understanding of neuropeptide functions in Alzheimer's and other neurodegenerative illnesses. The integration of the brain-gut axis, as well as the emphasis on prevention through lifestyle modifications, represent a contemporary, holistic approach to addressing neurodegeneration.

While we comprehend that a minor revision may be required to increase the clarity of the text, we are sure that the article can be improved in a timely way and will be ready for publishing after these minor revisions.

1.      Line no.  44–51. Make a clear distinction between the causes of sporadic and congenital (genetic) forms of misfolded proteins. By dividing it into smaller paragraphs, this part can be made easier to understand.

2.      Line no.  104-107: It indicate that only 1-5% of instances of EOAD are caused by familial Alzheimer's disease (FAD), which accounts for 15-20% of incidences. Make it clear if the 1-5% just relates to EOAD or if other forms of FAD are also included. Given that EOAD is a subset of FAD and that these percentages seem inconsistent, it could be confounding.

Neuropeptides, the modulators of different CNS processes

3.      What are the key differences between neuromodulation and neurotransmission?

4.      Include some brief comments regarding how particular neuropeptides may play roles in both the endocrine and central nervous systems.

GnRH

Line no. 443-445

5.      Some terminology (for example, "pulsatile GnRH release") are utilized without adequate explanation for readers unfamiliar with these ideas. It may be useful to quickly explain how pulsatile GnRH release differs from continuous release and why this distinction is important.

6.      If you can provide figures or flow diagrams demonstrating GnRH signaling pathways, its function in brain cancer, and its impact on neurodegeneration might help to improve knowledge and offer a visual overview of the essential aspects.

The hypocretins/orexins

7.      The section claims that orexin antagonists may provide both palliative and causal therapeutic options for Alzheimer's disease, however it would be helpful to provide more evidence or clinical trial examples to support up this claim.

Author Response

First we would like to express our gratitude for the constructive criticisms of the referee.

The manuscript title “The Aggravating Role of Failing Neuropeptide Networks in the Development of Sporadic Alzheimer's Disease” by Miklós Jászberényi et all, stands out for its clarity, breadth of study, and potential to considerably advance our understanding of neuropeptide functions in Alzheimer's and other neurodegenerative illnesses. The integration of the brain-gut axis, as well as the emphasis on prevention through lifestyle modifications, represent a contemporary, holistic approach to addressing neurodegeneration.

While we comprehend that a minor revision may be required to increase the clarity of the text, we are sure that the article can be improved in a timely way and will be ready for publishing after these minor revisions.

  1. Line no.  44–51. Make a clear distinction between the causes of sporadic and congenital (genetic) forms of misfolded proteins. By dividing it into smaller paragraphs, this part can be made easier to understand.

The referee is absolutely right. We made a clear distinction and changed the numbers accordingly

  1. Line no.  104-107: It indicate that only 1-5% of instances of EOAD are caused by familial Alzheimer's disease (FAD), which accounts for 15-20% of incidences. Make it clear if the 1-5% just relates to EOAD or if other forms of FAD are also included. Given that EOAD is a subset of FAD and that these percentages seem inconsistent, it could be confounding.

The referee is absolutely right. We made a clear distinction and changed the numbers accordingly

Neuropeptides, the modulators of different CNS processes

  1. What are the key differences between neuromodulation and neurotransmission?

We devoted a paragraph to the question

  1. Include some brief comments regarding how particular neuropeptides may play roles in both the endocrine and central nervous systems.

We added another paragraph to answer this intriguing question

GnRH

Line no. 443-445

  1. Some terminology (for example, "pulsatile GnRH release") are utilized without adequate explanation for readers unfamiliar with these ideas. It may be useful to quickly explain how pulsatile GnRH release differs from continuous release and why this distinction is important.

The referee is absolutely right. We added a section to explain this phenomenon 

  1. If you can provide figures or flow diagrams demonstrating GnRH signaling pathways, its function in brain cancer, and its impact on neurodegeneration might help to improve knowledge and offer a visual overview of the essential aspects.

Thank you for the suggestion. We added a new figure to outline our hypothesis.

The hypocretins/orexins

  1. The section claims that orexin antagonists may provide both palliative and causal therapeutic options for Alzheimer's disease, however it would be helpful to provide more evidence or clinical trial examples to support up this claim.

We think that in Alzheimer's disease the most important is prevention, which can be achieved by prevention. Restoration of healthy lifestyle and sleeping patterns are of utmost importance. Orexin antagonist may play a crucial role in this facilitating the glymphatic circulation. Therefore, we expanded the criticized section.